# Online Control for Biped Robot with Incremental Learning Mechanism

**Liang Yang** [1,2,*] **, Guanyu Lai** [3] **, Yong Chen** [2] **and Zhihui Guo** [1]

1   Zhongshan Institute, School of Computer Science, University of Electronic Science and Technology of China, Zhongshan 528402, China; dance007@126.com
2   School of Automation Engineering, University of Electronic Science and Technology of China, Chengdu 611731, China; ychencd@uestc.edu.cn
3   School of Automation, Guangdong University of Technology, Guangzhou 510006, China; lgy124@gdut.edu.cn
*   Correspondence: alex_yangliang@foxmail.com

**Abstract:** In this paper, we develop a new online walking controller for biped robots, which integrates a neural-network estimator and an incremental learning mechanism to improve the control performance in dynamic environment. With the aid of an iteration algorithm for updating, some newly incoming data can be used straightforwardly to update into the original well-trained model, in order to avoid a time-consuming retraining procedure. On the other hand, how to maintain the zero-moment-point stability and counteract the effect of yaw moment simultaneously is also a key technical problem to be addressed. To this end, an interval type-2 fuzzy weight identifier is newly developed, which assigns weight for each walking sample to deal with the imbalanced distribution problem of training data. The effectiveness of the proposed control scheme has been verified through a full-dynamics simulation and a practical robot experiment.

**Keywords:** biped robot; zero-moment-point; neural networks incremental learning

## 1. Introduction

In recent years, biped robot has received considerable attention owing to its unique bipedal movement, excellent suitability to human society and theoretical importance. Up to date, a number of active control approaches have been proposed, e.g., stability-criterion-based method [1–3], model-based method [4–6], and optimization-based method [7–9]. In addition to these, many real robot platforms have been successfully developed, including Atlas, MABEL, ASIMO, and NAO [10,11]. To achieve the bipedal locomotion stability, zero-moment-point (ZMP) was proposed and have become the most popular stability criteria. In [12], Kajita et al. designed a ZMP tracking servo controller and proposed a bipedal walking pattern generation algorithm based on cart-table model. In [13], a modified walking pattern method was presented by utilizing allowable ZMP variation and both step length and walking period can be independently adjusted without any extra step. Subsequently, Shin et al. [14] further proposed a practical gait synthesis algorithm by optimizing gait parameters, and the locomotion stability was guaranteed. Moreover, Caron et al. [15] defined the pendular support area and presented a whole-body controller for locomotion across arbitrary multicontact stances. Despite these contributions, the stability established in [12–15] depends on an assumption that the effect on stability caused by yaw moment can be ignored, which is in fact a restrictive condition. As pointed out in [16–18], yaw moment is inevitably generated by the motion of swing leg, which may lead to slippage or falling down.

To remove such limitation, much effort has been paid in this field and some interesting results were reported in [16–21]. In particular, Hirabayashi et al. [16] proposed a waist-rotation-based yaw moment compensation algorithm, while a biped robot was modeled as a 3D inverted pendulum. In [22], with the fusion of waist joint control technique

and optimized swing leg reference generation method, a fast walking pattern generation approach was presented to counteract the effect of yaw moment. Inspired by human walking experience, Xing et al. [18] designed an arm-swing-based control scheme to cancel the factors which produce the yaw moment. To further improve the control performance, in [19], the angular momentum rate changes were smoothly integrated into yaw moment equation and the locomotion stability was ensured by utilizing a Eulerian ZMP resolution approach. Moreover, Yang et al. [21] constructed a practical control scheme to compensate yaw moment by controlling lower limb. Although much progress has been made in dynamic balance control field, some challenging difficulties still remain open. In most of existing control schemes on yaw moment compensation for biped robot, such as those mentioned above, only a few joints are involved which bring much burden to driving motors and may result in unnatural gaits. In practice, it is difficult to generate natural and efficient gaits in real time according to external disturbance from circumstance.

To address this problem, a series of optimization-based methods were proposed. In [23], a spline-based estimation of distribution algorithm was proposed by formulating the gait pattern generation into a multiobjective optimization problem. In [24], besides ensuring the ZMP stability, the performance of energy efficiency was also well guaranteed by the fusion of moving ZMP criterion with the fourier series approximation technique. With the recourse of Newton–Raphson iteration, the locomotion stability was achieved and the walking speed of robot was successfully regulated online in [25]. Furthermore, Wang et al. [26] presented a SVM-based learning control system for biped robots, in which a novel SVM objective function with energy-related slack variables was proposed. This objective function followed the principle that the slack variables were determined by energy cost, which means the sample with lower energy consumption contributes more to SVM regression. This provided an interesting clue to learn biped walking locomotion. However, this method generally depends on a well-trained model, which may not always be achieved in practical applications.

Motivated by such an observation, in this paper, we make an attempt to further address the online control problem for biped robot. To remove the restrictions just mentioned, the main challenging difficulty that obstructs the design of our control scheme lies in the development of a protocol to compensate yaw moment and at the same time maintain zero-moment-point stability. To overcome the difficulty, an online walking control approach is presented. In summary, the work of this paper has the following novelties and contributions:

1. As compared with the control scheme developed in [26], ours newly equips with a neural-network estimator and an incremental mechanism, with which those newly coming data can be used straightforwardly to update the original well-trained model in real time. This implies that it is possible for a robot to achieve better locomotion stability in dynamic environment, e.g., from flat ground to uneven terrain;
2. Traditional optimization-based methods, such as those in [23–26] are involved in many adaption laws to be updated or computed online, which may result in a computation burden during control implementation. To remove this restriction, we achieve the fusion of the random vector functional-link neural network with an incremental mechanism, so that the entire retraining from beginning can be effectively avoided. Furthermore, by designing an interval type-2 fuzzy weight identifier (IT2FWI), both horizontal and vertical locomotion stabilities are successfully taken into account in training procedure.

The rest of this paper is organized as follows. In Section 2, the kinematics and dynamics of the biped robot are given and some preliminaries are presented. In Section 3, we propose an online control scheme based on incremental learning Algorithm 1, and a neural-network mechanism is established. In Section 4, simulation and experiment are carried out to verify the effectiveness of our scheme. In Section 5, the conclusions are given.

---

**Algorithm 1** Online Updating with increment learning algorithm

---

**Input:**
    incoming new samples $\mathbf{Q}_i = (\mathbf{q}_i, \dot{\mathbf{q}}_i, \ddot{\mathbf{q}}_i, \Delta X_z^i, M_z^i)$;
**Output:**
    $\mathbf{T} = (t_1, ..., t_N) = (\Delta \mathbf{q}_1, ..., \Delta \mathbf{q}_N)$
 1: Randomly initiate $\mathbf{W}_e, \mathbf{W}_h, \beta_e, \beta_h$, set training error $e = 0$;
 2: Calculate the matrix $\mathbf{A} = [\mathbf{K}^n \ \mathbf{H}^m]$;
 3: Calculate $(\mathbf{A}^m)^+$ with Equation (11);
 4: **while** $e \leq threshold$; **do**
 5:     Randomly initiate $\mathbf{W}_p, \beta_p$;
 6:     Set $\mathbf{Y}_p = \varphi(\mathbf{Q}W_p + \beta_p)$ and $\mathbf{A}^{m+1} = [\mathbf{A}^m | \mathbf{Y}^p]$
 7:     Calculate $(\mathbf{A}^{m+1})^+$ and $\mathbf{W}^{m+1}$ by Equations (12)–(14);
 8: **end while**
 9: Calculate $\mathbf{T} = (\mathbf{A}^{m+1})^+ \mathbf{W}^{m+1}$

---

## 2. System Description and Some Preliminaries

### 2.1. Overview of Biped Robot BRZ-4

BRZ-4 is a half-size biped robot, which is set up as a test bed, as presented in Figure 1. Basically, BRZ-4 is 66.2 cm in height and 2.4 kg in weight, which contains 17 degrees. Specifically, three degrees for hip joint, one for knee joint, and two for ankle joint. To collect necessary feedback information, each joint is driven by a DYNAMIXEL MX-64-T motor and the mechanical structure of this robot is made from 3D printing. Moreover, the rotary encoders are integrated in the motors to obtain the motion of joint. The BRZ-4's kinematic model is accordingly depicted in Figure 1 and the physical parameters are given in Table 1.

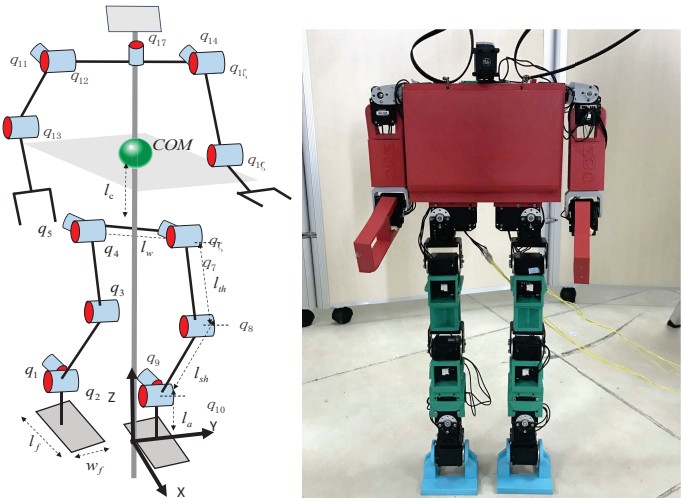

**Figure 1.** Kinematic model for BRZ-4.

**Table 1.** Basic Physical Parameters of BRZ-4.

|       | Link        | Trunk | Thigh | Shank | Arm   | Foot  |
|-------|-------------|-------|-------|-------|-------|-------|
| BRZ-4 | Length (cm) | 20    | 17.5  | 15.5  | 24.7  | 5     |
|       | Mass (kg)   | 1.05  | 0.256 | 0.156 | 0.156 | 0.075 |

A control system for BRZ-4 is set up to perform stable walking objective, which consists of biped robot and ground workstation. By integrating Matlab in the ground workstation, the proposed method is implemented to collect the real-time motion of robot and send control signals to BRZ-4 through RS485 bus. Moreover, to reduce the number of cables, driving motors are connected in daisy chains.

### 2.2. Kinematics and Dynamics

Usually, biped robot can be simplified as connective link model as shown in Figure 1, in which each link is uniform mass distribution. The relationship between joint speed and end-effector velocity is defined as

$$\dot{\mathbf{r}} = J(\mathbf{q})\dot{\mathbf{q}} \tag{1}$$

where $\dot{\mathbf{r}} \in R^m$ is task-space velocity, $\mathbf{q} = [q_1, ..., q_n] \in R^n$ represents joint angles, $n$ is the number of degrees of freedom, $\dot{\mathbf{q}} = [\dot{q}_1, ..., \dot{q}_n]$ denotes joint angle velocities and $J(\mathbf{q}) \in R^{m \times n}$ is the Jacobian matrix from joint space to task space.

From the Lagrangian approach, the dynamics of biped robot can be expressed as follows

$$M(\mathbf{q})\ddot{\mathbf{q}} + C(\mathbf{q}, \dot{\mathbf{q}})\dot{\mathbf{q}} + G(\mathbf{q}) + \mathbf{F}_e = \tau \tag{2}$$

where $M(\mathbf{q}) \in R^{n \times n}$ is the positive definite inertial matrix, $C(\mathbf{q}, \dot{\mathbf{q}}) \in R^{n \times n}$ is the Coriolis and centrifugal matrix, $G(\mathbf{q}) \in R^n$ is the gravitational force, $\mathbf{F}_e \in R^n$ denotes the external disturbance, $\tau \in R^n$ is the joint torques.

### 2.3. Bipedal Locomotion Stability

Zero-moment-point (ZMP) stability criterion is one of the most popular stability criteria, which is successfully applied in real biped robot platforms including ASIMO, NAO, and HRP. According to the definition of ZMP, zero-moment-point is the point on the sole, in which the horizontal component of the net moment caused by inertial and gravity forces is zero as shown in Figure 2.

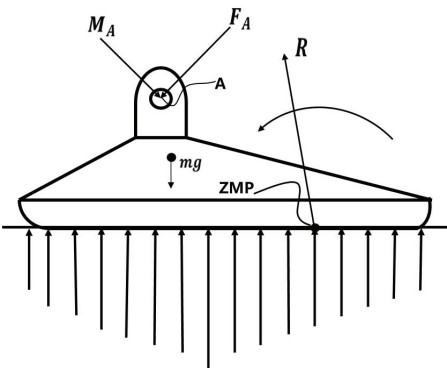

**Figure 2.** Forces acting on the supporting foot.

Hence, the following equation holds

$$M_x = M_y = 0 \tag{3}$$

where $M_x$ and $M_y$ are the x-axis and y-axis moment of the inertial and gravity forces, respectively.

As pointed out in [18,19], ZMP stability criterion cannot guarantee the moment equilibrium in vertical plane, which neglects the influence caused by yaw moment on locomotion stability. Actually, the undesired yaw moment along the support leg would be generated by the motions of components of biped robot in different planes. Thus, we have

$$M_z \leq M_R \tag{4}$$

where $M_z$ denotes yaw moment and $M_R$ is the moment generated by the ground reaction force. Specially, yaw moment is defined as below

$$M_z = \sum_{i=1}^{n} m_i(\mathbf{r}_i - \mathbf{r}_{zmp}) \times (\ddot{\mathbf{r}}_i + \mathbf{g}) \tag{5}$$

where $m_i$ is the mass of the $i$th connective link; $\mathbf{r}_i$ is the position vector of the center of the $i$th connective link; $\mathbf{r}_{zmp}$ denotes ZMP position vector; $\mathbf{g} = [0,0,g]$ is the gravitational acceleration vector; $M_z$ is yaw moment.

Moreover, ZMP coordinate $\mathbf{r}_{zmp} = (x_{zmp}, y_{zmp}, 0)$ has the following forms [27]:

$$x_{zmp} = \frac{\sum\limits_{i=1}^{n} x_i(\ddot{z}_i + g)m_i - \sum\limits_{i=1}^{n} z_i m_i \ddot{x}_i}{\sum\limits_{i=1}^{n} (\ddot{z}_i + g)m_i} \tag{6}$$

$$y_{zmp} = \frac{\sum\limits_{i=1}^{n} y_i(\ddot{z}_i + g)m_i - \sum\limits_{i=1}^{n} z_i m_i \ddot{y}_i}{\sum\limits_{i=1}^{n} (\ddot{z}_i + g)m_i} \tag{7}$$

where $[x_i, y_i, z_i]$ is the position of the center of the $i$th link.

To evaluate the locomotion stability in horizontal plane, ZMP stability margins introduced in [28] are adopted.

$$J_{zmp} = \begin{cases} \dfrac{1}{2}\dfrac{l_{zx}}{l_{cx}} + \dfrac{1}{2}\dfrac{l_{zy}}{d_{cy}} & if\ \mathbf{r}_{zmp} \in \Omega_{zmp} \\[2mm] 0 & if\ \mathbf{r}_{zmp} \notin \Omega_{zmp} \end{cases} \tag{8}$$

where $l_{zx}$ and $l_{zy}$ denote the x-axis and y-axis distance between zero-moment-point and boundaries, respectively; $l_{cx}$ and $d_{cy}$ represent the x-axis and y-axis distance between the center of foot sole and boundaries, respectively; $\Omega_{zmp}$ denotes ZMP boundaries.

## 3. Online Control System Design

In this section, the control system design procedure will be specially introduced and a walking control framework is presented as shown in Figure 3, in which the random vector function-link neural networks (RVFLNNs) [29] is adopted to approximate $f(\bullet)$ and an incremental learning mechanism is incorporated in the NNs. Moreover, an interval type-2 fuzzy weight identifier (IT2FWI) is designed to improve the control performance.

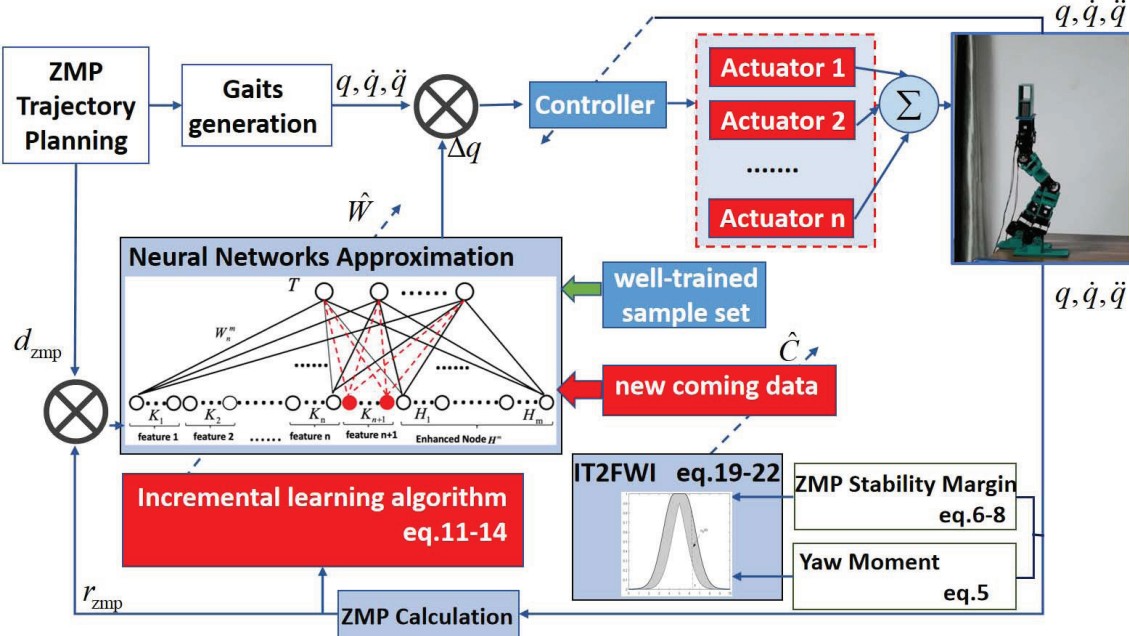

**Figure 3.** Architecture of proposed control system for biped robot.

### 3.1. Weighted Neural-Network Estimator

From Equations (5)–(7), it is noted that both ZMP stability and yaw moment are related with the position, velocity and acceleration of each link. According to [24,25] and results in [28], the locomotion stability can be ensured by regulating robot joints. Thus, the following mapping function is considered

$$\Delta \mathbf{q} = f(\mathbf{q}, \dot{\mathbf{q}}, \ddot{\mathbf{q}}, \Delta X_z, M_z) \tag{9}$$

where $f(\bullet)$ is a non-linear mapping function, $\Delta \mathbf{q} = [\Delta q_1, ..., \Delta q_n]^T$ denotes the corrections of all joints; $\Delta X_z$ is ZMP error, $M_z$ denotes yaw moment. By approximating the non-linear mapping function $f(\bullet)$, $\Delta \mathbf{q}$ can be obtained.

In our scheme, the random vector functional-link neural networks(RVFLNNs) [29] is adopted to estimate $f(\bullet)$. Different from traditional Neural Networks, RVFLNNs effectively eliminates the disadvantage of the long training process and provides a fast learning property by designing a flatted network with randomly generated weights and biases.

Given N training sample sequences $\{\mathbf{X}_i, t_i\}_i^N$, let $\hat{c}_i$ represent the appropriate weight of the *i*th sample. Thus, the approximation task is formulated as the following optimization problem

$$\arg \min_W : \sum_{i=1}^N \hat{c}_i (A_i W_i - T_i) + \lambda \|\mathbf{W}\|_2^2 \tag{10}$$

where $\hat{c}_i$ denotes the weight of the *i*th sample, $\mathbf{A} = [A_1, ..., A_N] = [\mathbf{K}^n \ \mathbf{H}^m]$ represents the input matrix, $\mathbf{K}^n = [\mathbf{K}_1, ..., \mathbf{K}_n]$ is the input node set, $\mathbf{K}_i = \varphi(\mathbf{X}\mathbf{W}_{ei} + \beta_{ei})$ is the input node, $\mathbf{X}$ is the input sample data, and $\mathbf{H}^m = [\mathbf{H}_1, ..., \mathbf{H}_m]$ is the enhancement node set, $\mathbf{H}_i = \varphi(\mathbf{K}^n \mathbf{W}_{hi} + \beta_{hi})$ is the enhancement node; $\beta_{ei}$ and $\beta_{hi}$ are bias; $\mathbf{W}_{ei}$ and $\mathbf{W}_{hi}$ are weight matrices; $\varphi(\bullet)$ is sigmoid function; $\mathbf{T} = [T_1, ..., T_N]$ is the desired output matrix, $\lambda$ is a penalty coefficient. Moreover, $\mathbf{W}$ is the connecting weight matrix, which can be computed by $\mathbf{W} = [W_1, ..., W_N] = \mathbf{A}^+ \mathbf{T}$.

By applying the Moore–Penrose inverse, we have

$$\mathbf{A}^+ = \lim_{\lambda \to 0} (\lambda \mathbf{I} + \hat{C} \mathbf{A} \mathbf{A}^T)^{-1} \mathbf{A}^T \hat{C}^T \tag{11}$$

where $\mathbf{A}^+$ is the pseudo-inverse matrix of $\mathbf{A}$, $\hat{C} = [\hat{c}_1, \hat{c}_2, ..., \hat{c}_n]$ is the weight matrix of sample data.

### 3.2. Incremental Learning Method Design

Let $\mathbf{A}^{m+1} = [\mathbf{A}^m \ \mathbf{Y}^p]$, then the pseudoinverse of $\mathbf{A}^{m+1}$ can be expressed as follows [29]:

$$(\mathbf{A}^{m+1})^+ = \begin{bmatrix} (\mathbf{A}^m)^+ - \mathbf{D}\mathbf{B}^T \\ \mathbf{B}^T \end{bmatrix} \tag{12}$$

where $\mathbf{A}^m = [\mathbf{A}_1, ..., \mathbf{A}_m]$ is the original input matrix, $\mathbf{Y}^p = [\mathbf{Y}_1, ..., \mathbf{Y}_p]$ denotes the new input node set, $\mathbf{Y}_i = \varphi(\mathbf{Q}\mathbf{W}_{p_i} + \beta_{p_i})$, $\mathbf{Q}$ is the new incoming data, $\mathbf{D} = (\mathbf{A}^m)^+ \mathbf{Y}^p$,

$$\mathbf{B}^T = \begin{cases} (\mathbf{C})^+ & , if \ \mathbf{C} \neq 0 \\ (1 + \mathbf{d}^T \mathbf{d})^{-1} \mathbf{B}^T (\mathbf{A}^m)^+, if \ \mathbf{C} = 0 \end{cases} \tag{13}$$

and $\mathbf{C} = \varphi(\mathbf{K}^n \mathbf{W}_p + \beta_p) - \mathbf{A}^m \mathbf{D}$.

Thus, the new weight matrix is achieved by the following equation:

$$\mathbf{W}^{m+1} = \begin{bmatrix} \mathbf{W}^m - \mathbf{D}\mathbf{B}^T \mathbf{T} \\ \mathbf{B}^T \mathbf{T} \end{bmatrix} \tag{14}$$



### 3.3. Interval Type-2 Fuzzy Identifier Design

One of the main difficulties in the development of the proposed control scheme is how to assign an appropriate learning weight for each walking sample. In this paper, an interval type-2 fuzzy weight identifier (IT2FWI) is designed to deal with the uncertainty of walking sample.

The interval type-2 fuzzy logic systems (FLSs) rules are given as follows

$$Rule\ l:\ \text{If } z \text{ is } \widetilde{B}_{l,1},\ y \text{ is } \widetilde{B}_{l,2},$$
$$\text{then } \hat{c} \text{ is } \widetilde{O}_l.$$

where $z$ and $y$ represent ZMP stability margin and yaw moment, $\hat{c}$ denotes the weight of sample data, $\widetilde{B}_{l,j}$ and $\widetilde{O}_l$ denote the linguistic variables of the fuzzy sets; $l = 1, 2, ..., L$. $L$ is the total number of the fuzzy rules.

Gaussian membership function is adopted to map crisp input to fuzzy sets for its clear physical signification. The membership function of ZMP stability margin is given as below

$$\phi_{\widetilde{B}_{i,j}}(z_i) = [\underline{\phi}_{\widetilde{B}_{i,j}}(z_i), \overline{\phi}_{\widetilde{B}_{i,j}}(z_i)] \tag{15}$$

$$\overline{\phi}_{\widetilde{B}_{i,j}}(z_i) = \begin{cases} (\underline{c}_{i,j}^z, \sigma_{i,j}, z_i) & , z_i < \underline{c}_{i,j}^z \\ 1 & , \underline{c}_{i,j}^z \leq z_i \leq \overline{c}_{i,j}^z \\ (\overline{c}_{i,j}^z, \sigma_{i,j}, z_i) & , z_i > \overline{c}_{i,j}^z \end{cases} \tag{16}$$

$$\underline{\phi}_{\widetilde{B}_{i,j}}(z_i) = \begin{cases} (\underline{c}_{i,j}^z, \sigma_{i,j}, z_i) & , z_i > \frac{\underline{c}_{i,j}^z + \overline{c}_{i,j}^z}{2} \\ (\overline{c}_{i,j}^z, \sigma_{i,j}, z_i) & , z_i \leq \frac{\underline{c}_{i,j}^z + \overline{c}_{i,j}^z}{2} \end{cases} \tag{17}$$

where $(c_{i,j}^z, \sigma_{i,j}, z_i) = \exp\{-\frac{1}{2}(\frac{z_i - c_{i,j}^z}{\sigma_{i,j}})^2\}$, and $c_{i,j}^z \in [\underline{c}_{i,j}^z \overline{c}_{i,j}^z]$.

From [30], the output fuzzy set $/_{\widetilde{O}(\hat{c})}$ can be obtained by the following equation

$$\phi_{\widetilde{O}}(\hat{c})$$
$$= \int_{\hat{c} \in [\underline{f}_1 \underline{\phi}_{\widetilde{G}_1}(\hat{c}) \vee \cdots \vee \underline{f}_L \underline{\phi}_{\widetilde{G}_L}(\hat{c}), \overline{f}_1 \overline{\phi}_{\widetilde{G}_1}(\hat{c}) \vee \cdots \vee \overline{f}_L \overline{\phi}_{\widetilde{G}_L}(\hat{c})]} \frac{1}{\hat{c}} \tag{18}$$

where $\underline{f}_i = \prod_{j=1}^{n} \underline{\phi}_{\widetilde{B}_{i,j}}, \overline{f}_i = \prod_{j=1}^{n} \overline{\phi}_{\widetilde{B}_{i,j}}, n = 2$; '$\vee$' operation denotes the maximum operation.

Utilizing the center-of-sets-type reduction and the Karnik–Mendel method, we have

$$\hat{c} = [\hat{c}_{low}, \hat{c}_{high}] \tag{19}$$

$$\hat{c}_{low} = \frac{\Sigma_{i=1}^{l}(\mathbb{Q}\overline{f})_i \widetilde{c}_i + \Sigma_{j=l+1}^{L}(\mathbb{Q}\underline{f})_j \widetilde{c}_j}{\Sigma_{i=1}^{l}(\mathbb{Q}\overline{f})_i + \Sigma_{j=l+1}^{L}(\mathbb{Q}\underline{f})_j} \tag{20}$$

$$\hat{c}_{high} = \frac{\Sigma_{i=1}^{r}(\mathbb{Q}\underline{f})_i \widetilde{c}_i + \Sigma_{j=r+1}^{L}(\mathbb{Q}\overline{f})_j \widetilde{c}_j}{\Sigma_{i=1}^{r}(\mathbb{Q}\underline{f})_i + \Sigma_{j=r+1}^{L}(\mathbb{Q}\overline{f})_j} \tag{21}$$

where $\hat{c}$ is an interval set, $\hat{c}_{low}$, $\hat{c}_{high}$ represent the left and right limits, respectively; $\hat{c}_i$ is the centroid of the type-2 interval consequent set $\widetilde{O}_i$; $\mathbb{C} = (\hat{c}_1, ..., \hat{c}_L)$ represent the original rule-ordered consequent values and $\widetilde{c} = (\widetilde{c}_1, \ldots, \widetilde{c}_L) = \mathbb{Q}\mathbb{C}$ satisfying $\widetilde{c}_1 \leq \widetilde{c}_2 \leq ...\widetilde{c}_L$; $\mathbb{Q}$ is an $L \times L$ permutation matrix.

Thus, the defuzzified output is

$$\hat{c} = \frac{\hat{c}_{low} + \hat{c}_{high}}{2} \tag{22}$$

## 4. Experiment Results and Analysis

In this section, the effectiveness of our control scheme is discussed through simulations and experiments. Robots are required to perform two typical kinds of tasks including walking on flat ground and climbing stairs. The first one is based on a physical platform, while the second one is carried out on a simulation platform.

### 4.1. Experiment: Walking on Flat Ground

As illustrated in Figure 1 and Table 1, the biped robot BRZ-4 is set up as test bed. Roughly, the test bed consists of two parts, which are the ground workstation and BRZ-4. We apply the proposed control algorithm to BRZ-4. Specifically, our scheme contains off-line and online learning parts. To improve the efficiency, the off-line training is carried out in matlab while the online learning is implemented by using C language. Moreover, the control commands and the states of robot can be transmitted through RS485 bus. To visually present the basic components of hardware system, a graphical result is provided in Figure 4.

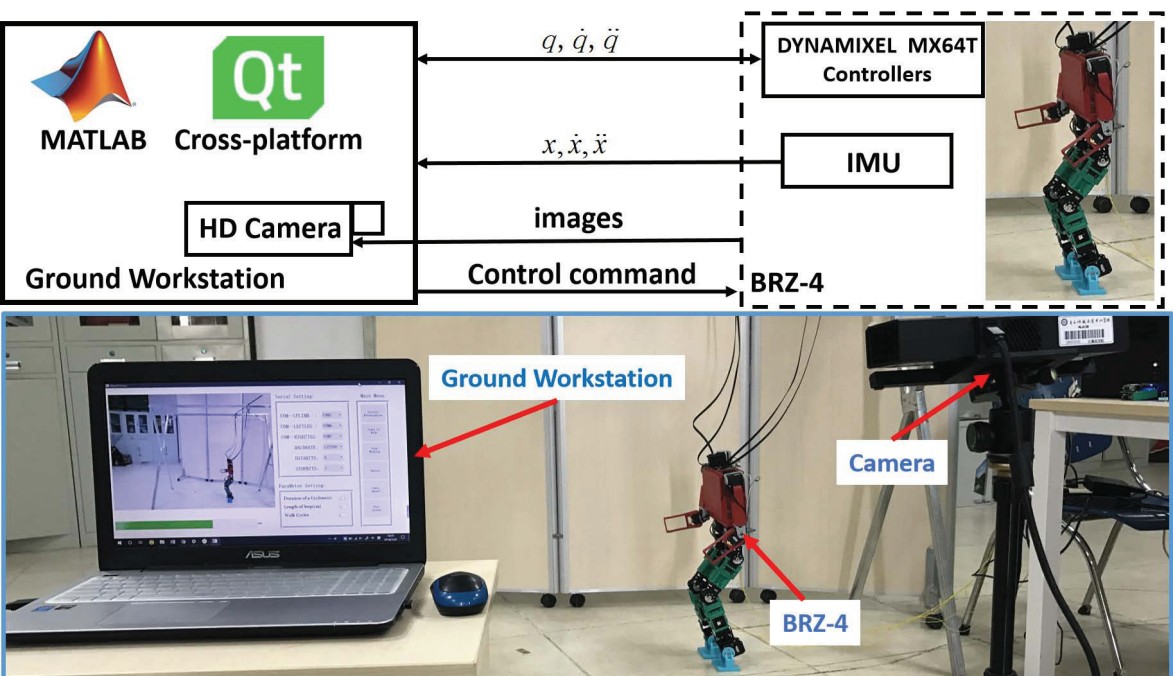

**Figure 4.** Basic components of hardware system.

One of the goals of the experiment is to control the biped robot to track the desired gait, such that stable walk is achieved. Under the new constructed control framework, desired gait is planned by using a spline-based parametric optimization technique [31], which contains start gait, period normal gait, and stop gait. In addition, the generation of planned gait is implemented in the ground workstation and specific control command will be transmit to BRZ-4 through RS485 bus in real time. To visually illustrate the planned results, the stick animation is presented in Figure 5, in which the CoM trajectory is highlighted in red.

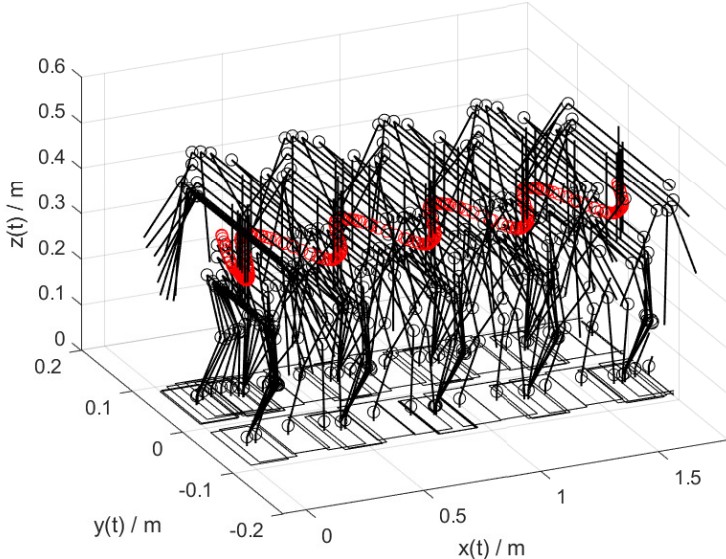

**Figure 5.** Animation of planned gait.

In the construction of the proposed IT2FWI, we take the Gaussian function with fixed standard deviation $\sigma$ and uncertain mean as the primary membership functions. By applying the trial-and-error procedure, the designed parameters of membership functions of ZMP stability margin and yaw moment are chosen as follows

$$\sigma_z = 0.12; \sigma_y = 0.18$$

$$[\underline{c}_{ij}^z, \bar{c}_{ij}^z]_{j=1}^5 = \{[-0.02, 0.02], [0.23, 0.27], [0.48, 0.52],$$
$$[0.73, 0.77], [0.98, 1.02]\}$$
$$[\underline{c}_{ij}^y, \bar{c}_{ij}^y]_{j=1}^5 = \{[-1.02, -0.98], [-0.52, -0.48], [-0.02, 0.02],$$
$$[0.48, 0.52], [0.98, 1.02]\}$$

A comparison between the proposed control scheme and the one in [26] is carried out on the platform of BRZ-4. The ZMP response trajectories are plotted in Figure 6. As indicated in this figure, all the ZMP trajectories are observed to be within the convex boundaries of the supporting foot, which implies that both methods can ensure the locomotion stability in horizontal direction. On the other hand, undesired yaw moment has an significant impact on locomotion stability of biped robot, as pointed out in [16–18]. Now we test the effectiveness of our control scheme in compensating for yaw moment. The evolutions of yaw moment, with our method and the one in [26] are visualized in Figure 7. As seen from the comparison, with the above two methods, the yaw moment is successfully suppressed. Apart from these, the root-mean-square (RMS) errors of x-axis/y-axis ZMP stabilities and yaw moment are recorded in Table 2. It is noted that, with the proposed scheme, the RMS errors of x-axis,y-axis ZMP trajectories and yaw moment are around 5.9%, 9.9%, and 20.7% lower than those in [26], respectively. Moreover, comparing with the method in [26], the online learning time is dramatically reduced from 1.56 s to 0.23 s.

**Table 2.** Comparison of control performance.

|  | **Proposed Method** | **Method in [26]** |
|---|---|---|
| RMS error ($x_{zmp}$) | 0.0322 | 0.0341 |
| RMS error ($y_{zmp}$) | 0.0251 | 0.0276 |
| RMS error ($M_z$) | 0.0617 | 0.0745 |
| Learning time of each cycle (s) | 0.23 | 1.56 |

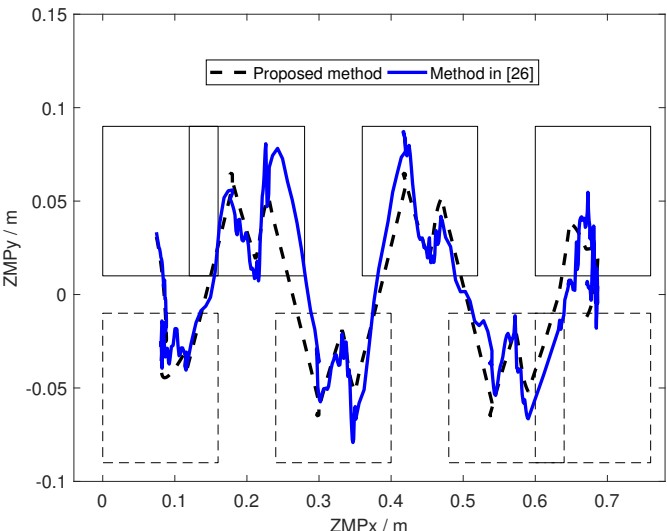

**Figure 6.** ZMP trajectories for BRZ-4 with two methods.

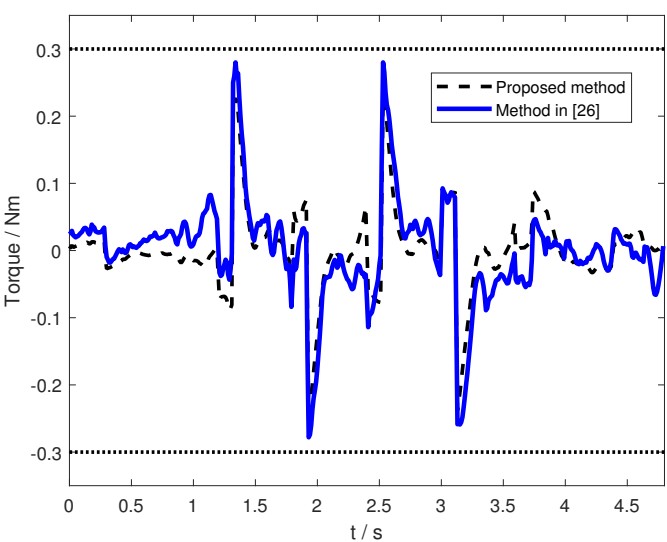

**Figure 7.** Yaw moment evolutions for BRZ-4 with two methods.

**Remark 1.** *Comparing with [26] which also focuses on the optimization-based learning control design, the learning mechanism in our scheme can be divided into two parts including off-line learning and online learning. By employing a flat network structure and deriving an weight matrix updating Equation (14), the proposed method successfully avoids the entire retraining from beginning. As a result, the online learning time is dramatically reduced as shown in Table 2.*

### 4.2. Simulation: Climbing Stairs

In this case, we consider the new robot BRZ-5, whose basic parameters Table 3. As indicated in Figure 8, BRZ5 is 121 cm in height and 14.9 kg in weight. The whole simulation contains two kinds of gaits. One is walking gait and the other is climbing stairs gait. Specially, every gait includes six step cycles. In this simulation, the robot is required to climb stairs after walking six steps on flat ground. Moreover, some comparative simulations between our proposed method with the one in [26] are conducted on Pybullet which is a real-time physics simulation platform. To facilitate the comparison and analysis, we keep the setting parameters and initial conditions as the same.

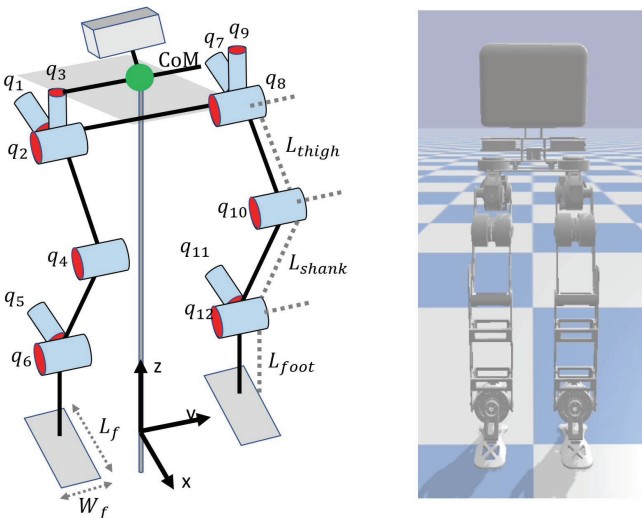

**Figure 8.** Kinematic model for BRZ-5.

**Table 3.** Basic physical parameters of BRZ-5.

|  | Link | Trunk | Thigh | Shank | Foot |
|---|---|---|---|---|---|
| BRZ-5 | Length (cm) | 38.3 | 37.1 | 35.0 | 11.0 |
|  | Mass (kg) | 1.38 | 3.58 | 2.18 | 1.048 |

The comparative simulation results are given in Figures 9 and 10. In particular, Figure 9 shows the snapshots of climbing stairs while the ZMP response trajectories are plotted in Figure 10. From Figure 9, it is noticed that, with these two methods, the robot can maintain balance in the first six step cycles on flat ground. However, in the next six step cycles, the robot fell down with the method in [26] while the robot controlled by the proposed scheme successfully finished the climbing stairs task. Similar results are also observed in Figure 10, in which ZMP response trajectories are illustrated in dot-dash black line and solid blue line. As indicated in Figure 10, with the approach in [26], ZMP response trajectory is basically within ZMP boundaries in the first six steps while the obvious deviation appears from the 8th to 12th steps. Comparing with the control scheme in [26], ours exhibits a better generalization performance.

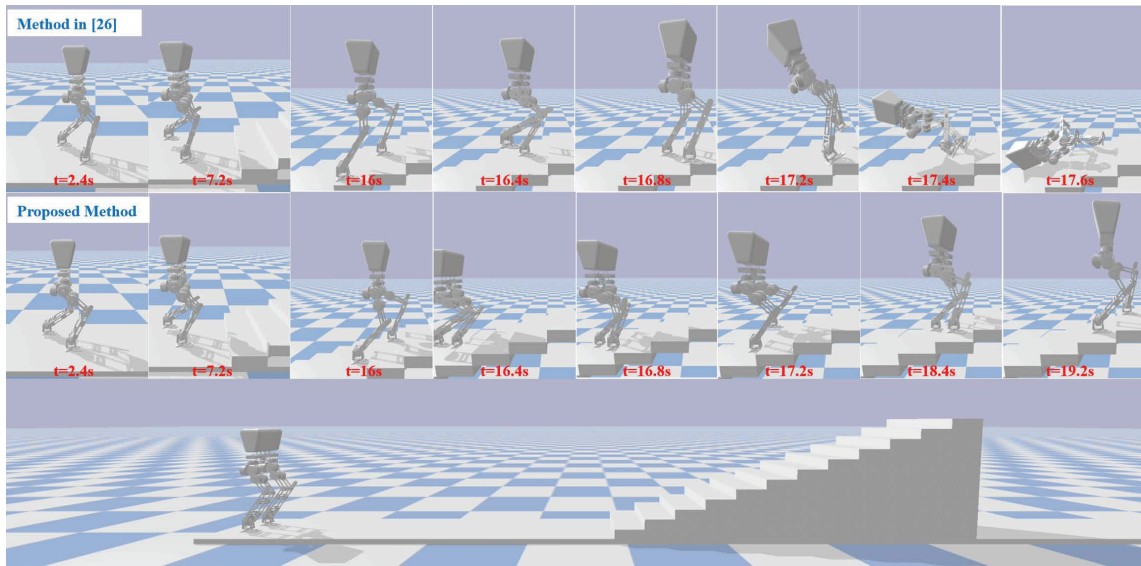

**Figure 9.** Snapshots of climbing stairs (Simulation platform: Pybullet).

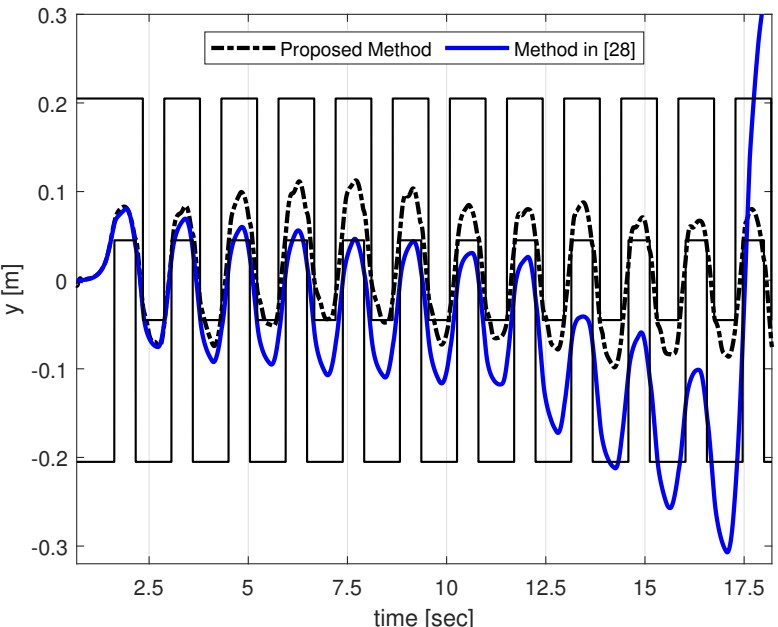

**Figure 10.** ZMP trajectories for BRZ-5 with two methods.

**Remark 2.** *A brief analysis is given to the above comparative results. As we know, the strong environment adaptive ability is one of the keys to realize the large-scale application of biped robots. However, it is almost impossible to handle all kinds of dynamic disturbances from environment with only one well-trained model. Unlike the approach in [26], an incremental updating mechanism is newly integrated into our scheme. With the aid of an iteration algorithm for updating, the new incoming data can be used straightforwardly to update into the original well-trained model, which successfully avoids the entire retraining from beginning. Thus, with this incremental updating mechanism, the adaptive capacity of robot is further improved.*

## 5. Conclusions

This paper presented a walking control framework for biped robot to deal with the online leaning and control problems. Under the new framework, an incremental learning algorithm is further constructed, such that the new coming data can be integrated into the well-trained model in real time without a retraining process. Finally, experiment and simulation results verified the effectiveness of the proposed scheme.

**Author Contributions:** Writing—original draft preparation, L.Y.; writing—review, G.L. and Y.C.; Simulation, Z.G. All authors have read and agreed to the published version of the manuscript.

**Funding:** The National Natural Science Foundation of China under Grant 61941301, 61803090, 11771102 and 61573108, in part by China Postdoctoral Science Foundation under Grant 2018M633353, in part by the Special Program for Key Field of Guangdong Colleges under Grant 2019KZDZX1037, in part by the Science and Technology Foundation of Guangdong Province under Grant 2021A0101180005 and 2019B090910001, in part by the Natural Science Foundation of Guangdong Province under Grant 2019A1515012109.

**Conflicts of Interest:** The authors declare no conflicts of interest.

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
