# Peer review of "Online Control for Biped Robot with Incremental Learning Mechanism"

_applsci, doi:10.3390/app11188599_

Round 1

Reviewer 1 Report

This paper presents an approach for an online control problem for biped robot. The motivation behind the proposed approach is clearly formulated, however the approach is not well explained. The experimental results are weak, as they only show a single case study compared with an approach of Wang et al. [26]. The discussion about the results is very short and not elaborated in enough detail. Please see below for more details.

Introduction: 
The authors write, that Wang et al. [26] provide "an interesting clue", without providing any details about the method. What kind of approach do they suggest? The authors also claim, that the method depends on a well-trained
model which is not always applicable in reality. However, the authors are doing the same with their approach, by taking also a well-trained model. Moreover, the authors write that Wang et al. take care of the uncertainity of training data. However, in the paragraph of the author's contribution in point 2. they claim, that most of the proposed approaches ignore it, omitting the approach where it has been done. It would much better to write how their own approach differs from Wang et al. In summary, it is not clear, what is the novelty and the contribution of the author's method compared to the existing literature. 

Authors explain the structure of their paper in the last paragraph. However, the sections  in the template of the paper are written in arabic numerals (2,3,4,5) and not in roman. The authors claim, that in section 2 a control problem is formulated, which is clerly not the case. 

Section 2:
Authors do not take care about correct mathematical notation. In eq. (1) r and q should be vectors and not scalars, which would be totally wrong. For matrices authors use the notation for sets. Moreover, no dimensions of the variables are given. The variable d is not explained. Eq. (5) has one bracket too much at the end of the equation. The gravitational acceleration
g is introduced twice, once before the eq (6) as gravity force, which is wrong and two equations  below correctly. This is really irritating. This is also the case for other variables. 

Section 3:
Authors have problems with punctuation: a dot is set right before the equation starts. There should be no dot, as a mathematical equation is part of a sentence. After equation (10) two lines of new variables follows, just seprated by comma withoud any explanation. This is also the case in the section 3.2, where only equations follow without any explanations.
fL from eq. (19) is not introduced. 

Section 4: 
The proposed approach is validated only on a single use case. 
Authors should carry out multiple experiments to show, that the method works and provides better results compared to the existing approaches. A critical discussion about the obtained results is missing completely. What are disadvantages of the method? When does it fail? 

Language/Grammar: Already the first sentence of the abstract contains a mistake "we presents". This happens in numerous cases throught the paper. 

Author Response

The authors would like to sincerely thank the reviewer again for sharing your thorough review and salient observations. Our response file has been attached.

Reviewer 2 Report

Fig 3. is of low quality.

Fig 4. is more confusing than helpful. The arrow directions are not meaningful.

MATLAB is notorious for being slow and inappropriate for real-time applications, yet, you use it. There is no discussion about execution times.

The results can be expanded to show more detail to draw more thorough conclusions.

There are some language errors.  

Author Response

Finally, the authors would like to sincerely thank the reviewer again for sharing your thorough review and salient observations. Our response file has been attached. Thank you.

Reviewer 3 Report

This is a good ms!

Author Response

The authors would like to sincerely thank the reviewer again for your valuable comment.